# Phase-separated droplets swim to their dissolution

Etienne Jambon-Puillet[1,2], Andrea Testa[1], Charlotta Lorenz[1,3], Robert W. Style [1], Aleksander A. Rebane[1,4] & Eric R. Dufresne [1,3] ✉

Biological macromolecules can condense into liquid domains. In cells, these condensates form membraneless organelles that can organize chemical reactions. However, little is known about the physical consequences of chemical activity in and around condensates. Working with model bovine serum albumin (BSA) condensates, we show that droplets swim along chemical gradients. Active BSA droplets loaded with urease swim toward each other. Passive BSA droplets show diverse responses to externally applied gradients of the enzyme's substrate and products. In all these cases, droplets swim toward solvent conditions that favor their dissolution. We call this behavior "dialytaxis", and expect it to be generic, as conditions which favor dissolution typically reduce interfacial tension, whose gradients are well-known to drive droplet motion through the Marangoni effect. These results could potentially suggest alternative physical mechanisms for active transport in living cells, and may enable the design of fluid micro-robots.

Cells control vital bio-chemical processes by creating compartments with distinct compositions. While classical organelles are enclosed by lipid membranes, membraneless organelles hold themselves together[1,2]. The absence of a membrane facilitates their condensation and dissolution[3], leads to significant interfacial tensions that drive capillary phenomena[4–6], and enables passive diffusion of solute across their surface[7]. Over the last decade, there has been tremendous progress toward understanding the chemical and physical processes that underlie the condensation of biological macromolecules[8–13].

Even though biochemical activity is thought to lay at the heart of membraneless organelle physiology, little is known about activity's impact on condensate stability, properties, and behavior. Experiments in living cells have shown that the fluid properties of some membraneless organelles require ongoing energy consumption[14–16]. Recent in vitro experiments have shown that chemical reactions can drive condensation[17] and suppress Ostwald ripening[18]. Our previous work[19] shows that enzymatic activity can generate fluid flow within model condensates. Theoretical studies predict that chemical activity can regulate condensation and dissolution[20], trigger division[21], and drive chemophoretic motility[22].

Here, we show that biomolecular condensates can swim toward solvent conditions that favor their dissolution. This process, which we call "dialytaxis", originates from a generic coupling of phase equilibria and interfacial properties. It emerges naturally whenever droplets, or their surroundings, generate chemical gradients. This simple physical process suggests alternative mechanisms for active transport within cells, and could enable the design of microscopic fluid robots.

## Results

### Active BSA droplets swim toward each other

Our model system is schematized in Fig. 1a. We trigger liquid-liquid phase separation by mixing a relatively dilute protein solution (bovine serum albumin (BSA) in PBS buffer (pH 7)) with a concentrated smaller neutral polymer (PEG 4k) (see Methods - condensate preparation). Because this condensation is driven by non-specific depletion interactions[23], additional proteins, such as enzymes, readily partition into the BSA-rich phase[19]. When the substrate of a partitioned enzyme is available in the dilute phase, these artificial condensates act as chemical micro-reactors. BSA droplets that partition urease catalyze the conversion of urea to ammonia and carbon dioxide:

[1]Department of Materials, ETH Zürich, Zürich, Switzerland. [2]LadHyX, CNRS, Ecole Polytechnique, Institut Polytechnique de Paris, Palaiseau, France. [3]Department of Materials Science and Engineering, Department of Physics, Cornell University, Ithaca, NY, USA. [4]Life Molecules and Materials Lab, New York University Abu Dhabi, Abu Dhabi, United Arab Emirates. ✉e-mail: eric.r.dufresne@cornell.edu

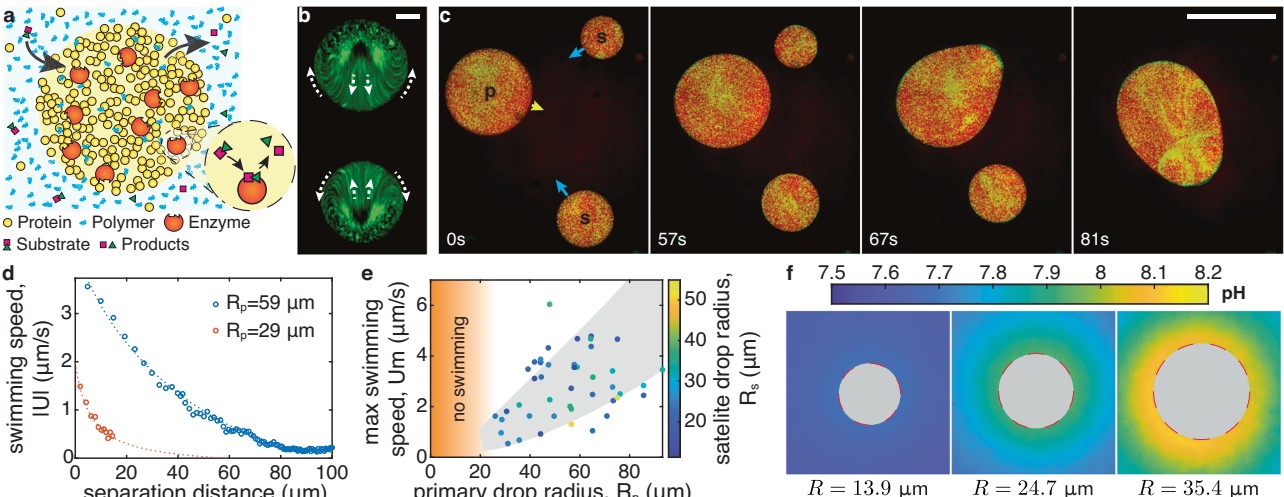

**Fig. 1 | Chemically active protein condensates swim toward each other.**
**a** Schematic of a chemically active protein condensate. Polymers act as depletants, triggering condensation. Enzyme-rich droplets act as micro-chemical reactors. **b** Time projection over 1 min of fluorescent particles inside two adjacent chemically-active protein droplets catalyzing the urea-urease reaction (overall enzyme concentration $c_e = 0.6$ μM, substrate concentration $c_s = 100$ mM). The droplets are pinned on the surface (PEGDA 700). Arrows indicate the internal flow direction. Scale bar, 10 μm. **c** Image sequence of chemically active droplets on a non-wetting surface (PEGDA 12k gel, $c_e = 1.2$ μM, $c_s = 100$ mM, see Movie S1). The primary (p), satellite drops (s) and direction of motion are indicated in the first panel. Scale bar, 100 μm. **d** Swimming speed of two satellite drops, $|U|$, as a function of the distance from their primary drops of radii $R_p$. Dashed curves are guides to the eye. **e** Maximum swimming speed, $U_m$, as a function the primary drop radius, $R_p$, for 48 swimming-induced coalescence events ($c_e = 1.2$ μM, $c_s = 100$ mM). The color codes the satellite drop radius, $R_s$, the gray area is a guide to the eye. No swimming was observed for drops below 25 μm in radius as indicated by the orange area. **f** pH imaging around pinned reacting droplets of increasing size ($c_e = 1.2$ μM, $c_s = 100$ mM, PEGDA 700). The fluorescence signal inside the drops is masked by a gray disc, since they do not contain pH dye. Source data for (**d**, **e**) are provided as a Source Data file.

$$CO(NH_2)_2 + H_2O \xrightarrow{\text{urease}} 2NH_3 + CO_2 \text{ (see Methods - reaction diffusion}$$
model, and[19,24]).

Active urease-loaded droplets develop internal flows that drive collective motility. On a glass surface functionalized with polyethylene glycol diacrylate with $M_n = 700$ (PEGDA 700), active condensates develop internal flows oriented toward neighboring droplets, but remain pinned (Fig. 1b)[19]. Using a much larger molecular weight of $M_n = 12{,}000$ yields a swollen gel (PEGDA 12k) that substantially reduces the adhesion energy between droplets and surfaces (see refs. 25,26 and Methods - coatings preparation). On this surface, reacting condensates swim toward each other with speeds on the order of μm/s (Fig. 1c, Movie S1, and ref. 27).

Like motion driven by attractive potentials (e.g., gravity or electrostatics), the speed of the droplets depends on their size and separation. Small satellite drops accelerate as they approach a large primary drop (Fig. 1d). Larger primary droplets attract satellites over farther distances. The maximum swimming velocity of satellite drops increases with the radius of the primary droplet (Fig. 1e).

We suspected that these long-range interactions are driven by gradients in the concentration of the chemical species that are consumed and/or produced by the droplet-localized enzymatic reaction. While we could not visualize these small molecules directly, we could observe their downstream effect on the pH. Using the pH-sensitive dye HPTS (see Methods - pH measurements, and Fig. 5) on stationary droplets stuck on PEGDA 700, we found a cloud of increased pH in steady-state around each reacting condensate (Fig. 1f). Like the motion of satellite droplets, the pH gradient increases with primary droplet size, and is largest near its surface. These size dependent profiles result from the coupled reaction and diffusion of multiple species in and around our droplet micro-reactors as shown in Methods - reaction diffusion model and Fig. 6.

### Passive BSA droplets swim in gradient chambers
Each of the chemical species involved in the enzymatic reaction could affect droplet motion. To disentangle their contributions, we probed the response of passive (enzyme-free) condensates to controlled gradients in a simple microfluidic device (schematized in Fig. 2a). Enzyme-free condensates were loaded into the central bridge, which was connected to reservoirs of its coexisting dilute phase on one side and dilute phase containing an additional chemical on the other side (see Methods - gradient chamber experiments). For ammonia ($NH_3$ (aq)), the gradient could be visualized indirectly through its effect on pH, as shown in Fig. 5e, f. Passive droplets in the bridge show an internal flow directed toward the source of $NH_3$ (aq) (Fig. 2b), as seen with the active droplets. Suspecting that pH gradients drive motility, we observed the response of passive droplets to sodium hydroxide (NaOH). Again, droplet motion was directed toward increased pH (Fig. 2c). However, this pattern broke down when we explored the impact of the second enzymatic product, $CO_2$, which is transformed to bicarbonate in our experimental conditions. In that case, we observed flow directed toward lower pH, away from the slightly basic $NaHCO_3$ source at pH 7.9 (Fig. 2d). Finally, gradients of the enzymatic substrate, urea, do not drive measureable flow (see Movie S2).

### Solute-driven Marangoni flow
From a fluid mechanics perspective, solute gradients can generate motility through two known mechanisms[28]. They can modify the interfacial tension and drive droplet motion through Marangoni flow[29,30]. Alternatively, diffusiophoretic motion arises when the interaction of solute and droplet across a diffuse layer generates an apparent slip velocity[31,32]. While the two mechanisms can work together, Marangoni effects should dominate for micron-scale liquid droplets[28,33]. In that case, droplets swim down interfacial tension gradients and develop internal flows with a maximum speed

$$V_m^{th} = \Delta\gamma/6\eta. \tag{1}$$

Here, $\Delta\gamma$ is the difference in interfacial tension across the droplet and $\eta$ is the viscosity of the more viscous phase[34]. This result neglects the

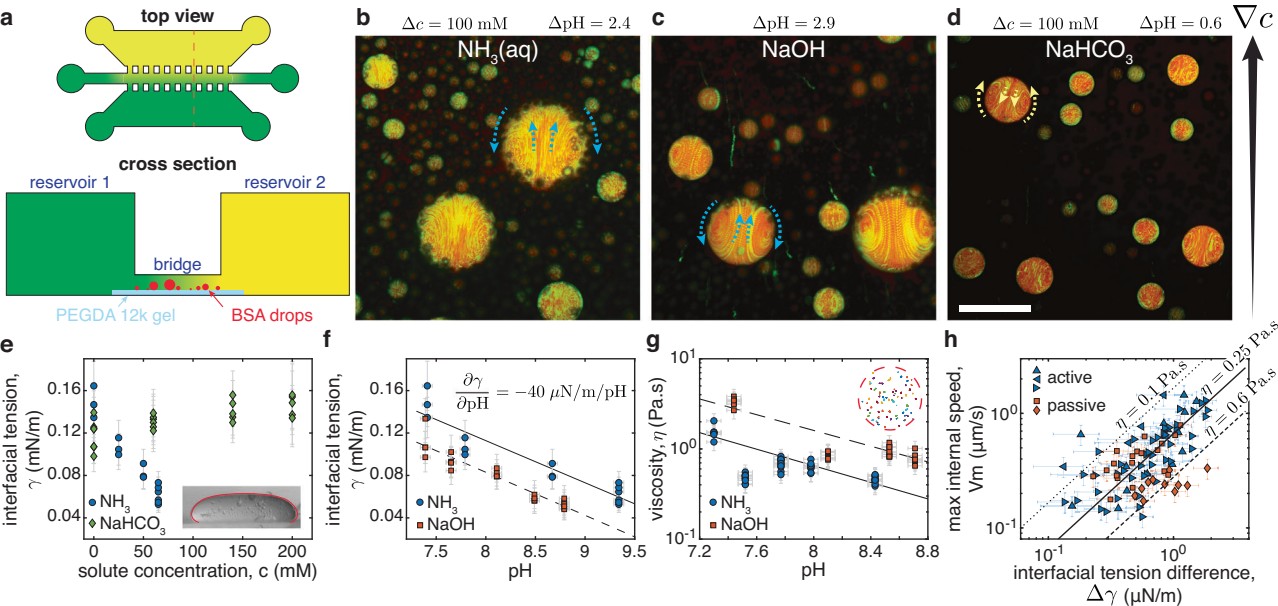

**Fig. 2 | Diverse solute gradients drive droplet motion. a** Schematic of the gradient chamber including a top view and a cross section (orange dashed line in top view). The first reservoir contains the dilute, BSA-poor, phase (colored green). The second reservoir is filled with dilute phase plus a solute (colored yellow). A narrow bridge coated with a PEGDA 12k gel and filled with coexisting dilute and dense phases connects the two reservoirs, allowing the solute to diffuse to produce a gradient around the droplets. **b–d** Time projection over 200 s of fluorescent particles (green) in passive BSA-rich condensates (red) inside the gradient chamber. The extra solutes, the concentration, and pH difference between reservoirs are indicated. The gradient direction is indicated by the black arrow and the internal flow direction by the dashed arrows (see also Movie S2). Drops swim towards the gradient for NH₃ (aq) and NaOH but away from it for NaHCO₃. Scale bar, 100 μm. **e** Equilibrium interfacial tension, γ, as a function of solute concentration c for NH₃ (aq) and NaHCO₃. Measured through the sessile drop method[59], the inset shows a typical drop flattened by gravity of width 1.8 mm and the Young-Laplace fit.

**f** Equilibrium interfacial tension, γ, as a function of pH when adding either NH₃ (aq) or NaOH. Lines are linear fits. **g** Droplet viscosity, η, as a function of pH for the same solutes, measured with passive microrheology. Lines are guides to the eye. The inset show particle tracks longer than 8 sec in a 39 μm drop. **h** Maximum internal velocity $V_m$, measured as a function of the interfacial tension difference across the drop $\Delta\gamma = 2R(d\gamma/dpH)(dpH/dx)$ for 93 individual droplets pinned on PEDGA 700, both chemically active (blue) and passive ones in a gradient of NH₃ (aq) (orange). Symbols differentiate experiments with independent condensate preparation. The straight lines show the predictions for idealized Marangoni swimmers of viscosities η = 0.1, 0.25, 0.6 Pa s. All points in (**e–h**) are individual measurements and all error bars convey non-statistical measurement uncertainties that account for the accuracy of the measuring instruments and measuring procedure (see Methods - interfacial tension measurement and - internal velocity measurement). Source data for (**e–h**) are provided as a Source Data file.

effects of confining surfaces and finite viscosity ratios, and assumes a linear gradient of interfacial tension.

The Marangoni mechanism explains the direction of flow in response to diverse solute gradients. The solutes NH₃ (aq), NaOH, and NaHCO₃ each impact the interfacial tension of passive condensates, as shown in Fig. 2e, f (see Methods - interfacial tension measurement, and Fig. 7). As suggested from Eq. (1) and the observed directions of swimming in Fig. 2, NH₃ (aq) and NaOH reduce the interfacial tension, while NaHCO₃ increases it.

The Marangoni mechanism predicts the observed flow speeds for active (urease-loaded) and passive droplets (in an ammonia gradient). We calculate the interfacial tension differences across droplets by combining pH measurements (see Methods - pH measurements, and Fig. 5d) on stationary droplets stuck on PEGDA 700 with the measured equilibrium interfacial tensions as function of pH for ammonia (Fig. 2f), $\Delta\gamma = 2R(d\gamma/dpH)(dpH/dx)$. We measure the maximum internal flow speed of the same stuck droplets with particle tracking velocimetry (see Methods- internal velocity measurement, and Fig. 8). We determine droplet viscosity in passive systems using particle tracking microrheology. Notably, we found droplet viscosities of the order of 1 Pa s that decayed exponentially with pH, by a factor of roughly three per pH unit (Fig. 2g). Combining these results, we plot $V_m$ against $\Delta\gamma$ for 93 individual droplets in Fig. 2h. A linear trend is clear for active droplets pooled across all experiments. For passive droplets, we see a clear trend within individual experiments. Predictions of Eq. (1) are in reasonable agreement with the data for a range of droplet viscosities from 0.1 to 0.6 Pa s. Condensate motility is thus driven by gradients of

solutes that affect their interfacial tension. The strong effffect of these solutes on the interfacial tension of our aqueous PEG-BSA system is, however, unexpected. They are not surfactants and barely affect the interfacial tension of water at these concentrations[35,36].

## Dialytaxis

The impact of enzymatic products on the interfacial tension emerges from their effect on phase coexistence. An important clue can be found in the behavior of large active droplets (R ≳ 50 μm), which tend to dissolve over long timescales (Movie S3). This suggests that high concentrations of reaction products can dissolve droplets. Indeed, macroscopic experiments with passive PEG-BSA solutions exhibit no phase separation when their pH is increased above about 9, either by the addition of NH₃ (aq) or NaOH. The coupling between droplet dissolution and motility is vividly demonstrated in Fig. 3a, b, and Movie S4. Here, we load one side of a gradient chamber with the dilute phase adjusted to pH 13.8 through the addition of NaOH, and image droplets as the sharp transient pH front moves across the field of view. As the pH front advances, an internal flow oriented toward the basic reservoir accelerates. Eventually, the droplets swim rapidly toward the basic reservoir and dissolve (Movie S4). Together, these observations suggest a direct link between interfacial tension and phase equilibria.

The interfacial tension of coexisting liquid phases depends on their compositions, which can be shifted by solutes. As the overall composition approaches the critical point, the two phases become more similar, and the interfacial tension vanishes[37–39]. The measured phase diagram of our PEG-BSA system is shown in Fig. 3c. Our overall

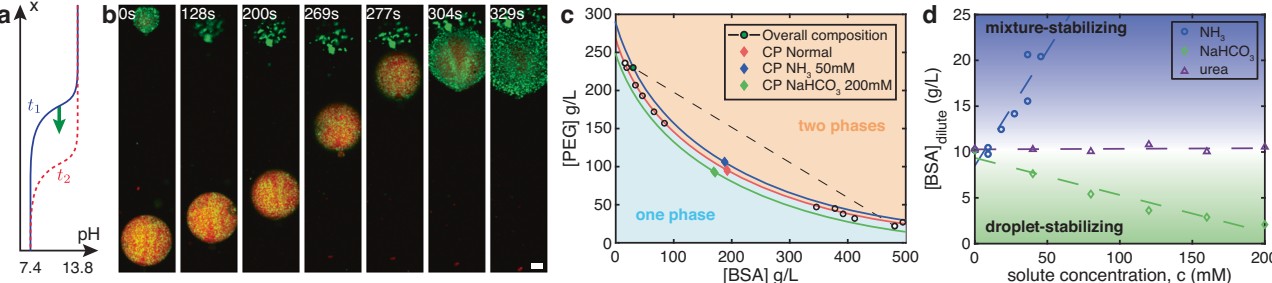

**Fig. 3 | Droplet motility driven by perturbations of phase equilibria.**
**a** Schematic of the sharp pH front propagating from top to bottom in (**b**). **b** Image sequence of passive condensates moving and dissolving in response to the sharp pH front. Scale bar, 10 μm (see also Movie S4, PEGDA 12k coating). **c** Phase diagram of the PEG-BSA system. In the absence of solute, circles show the measured binodal[19] and a red diamond shows the critical point. We prepare our droplets at the overall composition indicated by the green circle, and the system phase-separates along the associated tie line (dashed line and white circles). The background colors and colored curves are guides to the eye. Ezymatic products shift the critical point (blue 50 mM NH₃ (aq) and green 200 mM NaHCO₃ diamonds). **d** Shifts of BSA concentration in the dilute-phase as solutes NH₃ (aq), NaHCO₃ and urea are added at fixed overall BSA and PEG concentrations. Dashed lines are linear fits and shaded areas are guides to the eye. Source data for (**c, d**) are provided as a Source Data file.

composition (green circle) is close to the PEG-rich arm of the coexistence curve. The system phase separates along the dashed tie-line that connects the compositions of the two coexisting phases. The critical point (red diamond) shifts toward the overall composition with added NH₃ (aq) (blue diamond), shortening tie-lines, and lowering interfacial tension. NaHCO₃ shifts the critical point in the opposite direction, stabilizing the droplets and increasing their interfacial tension.

Established theories of polymer mixtures allow us to build a simple model linking interfacial tension and phase equilibria[40]. For segregative phase-separation, we expect $\gamma \sim (\phi_m/\phi_c - 1)^\nu$ where $\phi_c$ is the total concentration of macromolecules at the critical point, and $\phi_m$ is the concentration of macromolecules at the center of the tie-line[39]. Far from the critical point, mean-field theory predicts $\nu = 3/2$. Therefore, given an empirical shift of the critical point with solute concentration, $\partial \phi_c / \partial c$, we expect a corresponding shift in the interfacial tension, $d\gamma \sim -(\partial \phi_c/\partial c)dc$. In a solute gradient, the swim velocity will therefore scale like

$$U \sim \frac{R}{\eta} \frac{\partial \phi_c}{\partial c} \frac{\partial c}{\partial x}. \tag{2}$$

For $\partial \phi_c / \partial c > 0$, droplets will swim up the solute gradient. Conversely, droplets swim down the solute gradient when $\partial \phi_c / \partial c < 0$. This simple result captures essential features of our observations. While Eq. (2) was derived for segregative phase separation, we expect shifts of the critical point toward the overall composition to generally reduce interfacial tension. In other words, we expect droplets to swim to their dissolution. We call this process dialytaxis, from the Ancient Greek διάλυσις (dialysis), for dissolve, and τάξις, (taxis), for arrangement.

While the susceptibility of the critical point to a solute, $\partial \phi_c / \partial c$, may be challenging to predict from first principles, it can be efficiently measured. For macroscopically phase separated systems, the critical point can be found rapidly using the method of equal volumes (as in Fig. 3, see Methods - critical point measurement). However, this is impractical for most biologically relevant condensates, where only small volumes of condensed phases are available. In those cases, the phase behavior can be probed with microfluidics[41]. Alternately, we propose to simply measure dilute phase concentrations of droplet components as a function of solute concentration. Such a measurement can be performed optically (UV absorption or fluorescence) and requires only small amounts of protein. To demonstrate its feasibility, the concentration of BSA in the dilute phase of our system is plotted as a function of ammonia and bicarbonate in Fig. 3d. We clearly see that ammonia drives dissolution while bicarbonate stabilizes droplets,

consistent with the observed shifts in the critical point. As a control, we find that urea, whose gradients were not able to drive droplet motility, does not affect the stability of droplets. While a single measurement of the dilute phase concentration has less predictive power than directly measuring the interfacial tension or critical point, it is simple enough to screen large numbers of metabolites for their potential to drive droplet motion.

## Discussion

We have introduced dialytaxis in the context of macromolecular solutions without interfacially active chemical species. However, we note that it qualitatively describes the behavior of other two-phase systems. For example, oil droplets swim toward higher concentrations of ethanol, which promotes dissolution in ternary mixtures of oil, ethanol, and water[42]. Similarly, droplets dispersed in surfactant solutions swim spontaneously[33,43,44] toward empty micelles that enable their dissolution. Dialytaxis may provide a simple unifying framework for a wide range of solutal Marangoni effects. In the presence of surface active molecules, however, it may not have a dominant effect.

While simple, dialytaxis can drive a rich diversity of behaviors, summarized in Fig. 4. Droplets swim up gradients of solutes that promote mixing, and down gradients of solutes that promote condensation. Active droplets that produce droplet-destabilizing solutes swim toward one another, while active droplets that produce self-stabilizing solutes are expected to swim away from each other. The latter could potentially lead to highly organized steady states. Co-dispersions of active droplets with antagonistic effects on phase equilibria should display non-reciprocal interactions, including predator-prey dynamics[45–48].

Broadly speaking, dialytaxis suggests alternative mechanisms of active transport and mixing within living cells, independent of molecular motors or the cytoskeleton. The strength of dialytaxis compared to diffusion can be estimated using a Peclet number Pe = $UR/D$. Combining Eq. (1) with Stokes-Einstein relationship for the condensate diffusivity, we have Pe ≈ $(\eta_{dilute}/\eta)(\pi R^2 \Delta\gamma/kT)$. The surface tension difference $\Delta\gamma$ depends on the solute concentration gradient and how much it perturbs the phase equilibrium but is bounded by $\gamma$, the condensate interfacial tension. Condensate physical properties can vary by at least an order of magnitude depending on the condensate, and are often only measured for simplified in vitro systems. Nonetheless, assuming a moderate perturbation of the phase equilibrium $\Delta\gamma \sim 0.1\gamma$ and using typical values found in the literature[3–5,49,50] $10 < \eta/\eta_{dilute} < 100$, $0.1 < \gamma$ (μN/m) $< 100$, and $0.5 < R$ (μm) $< 10$ we find $10^{-2} < $ Pe $< 10^5$. This simple estimate suggests

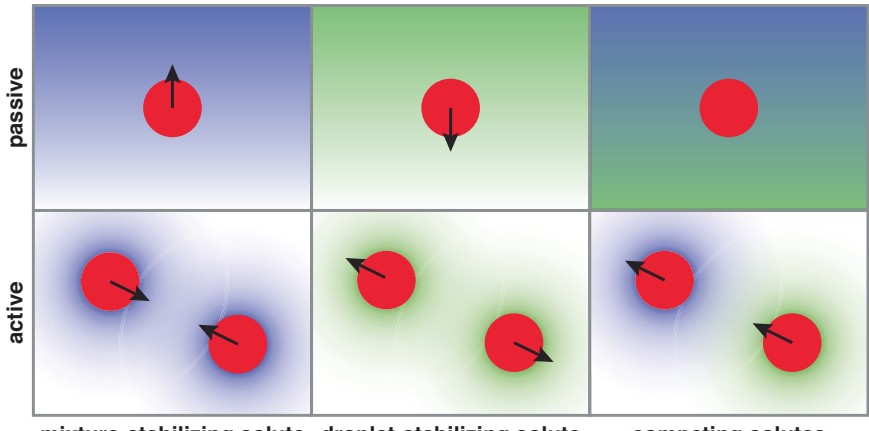

**Fig. 4 | Diverse responses to solubility-perturbing solutes.** (top) passive droplets are attracted to mixture-stabilizing solutes and repelled from droplet-stabilizing solutes. Opposing gradients can lead to no net motion. (bottom) Droplets producing mixture-stabilizing (resp. droplet-stabilizing) solutes swim toward (resp. away from) each other. Droplets producing competing solutes can chase each other.

that dialytaxis could be stronger than diffusion for a wide range of condensates, especially large ones, with small viscosity contrast, and a large interfacial tension.

We envision multiple contexts where dialytaxis could impact the physiology of membraneless organelles. Most simply, dialytaxis could regulate their coalescence[51]. Further, chemical gradients could contribute to assymmetric inheritence of membraneless organelles during cell division[3]. Similarly, dialytaxis could create symmetry-breaking fluxes of intermediaries for multi-step syntheses, such as ribosome biogenesis[4]. Finally, localized reactions could act as chemical beacons that bring biocondensates together to efficiently deliver diverse reactants.

Moreover, the generality and versatility of this simple chemo-mechanical coupling makes active macromolecular condensates an intriguing candidate platform for micro-robots[52,53]. The results in this manuscript already demonstrate rudimentary capabilities of sensing, actuation, and energy harvesting. Through a complex network of reactions within a single droplet, or through the interactions of diverse droplet types, active droplets should further be able to process information and implement control systems.

## Methods

### Materials
Polyethylene glycol 4000 Da (PEG 4k) (A16151), Polyethylene glycol 12 kDa (PEG 12k) (042635) were purchased from Alpha-Aesar. Bovine serum albumin (BSA) (A7638), Potassium phospate dibasic trihydrate (60349), Potassium chloride (KCl) (60128), 3-(Trimethoxysilyl)propylmethacrylate (440159), 2-Hydroxy-4'-(2-hydroxyyethoxy)-2-methylpropiophenone (410896), Polyethylene glycol diacrylate 700 Da (PEGDA 700) (455008), Jack bean urease (U4002), Rhodamine-B (R6626), 8-Hydroxypyrene-1,3,6-trisulfonic acid trisodium salt (HPTS) (H1529) were purchased from Sigma-Aldrich. Potassium phosphate monobasic (42420) was purchased from Acros Organics. Deuterium oxide (D$_2$O) (DE50B) was purchased from Apollo. N,N-dimethylformamide (DMF) (D119), Sodium bicarbonate (S/4240/53), Sodium hydroxide (S/4920) were purchased from Fisher Chemicals. Ammonia 25% (1133.1000), Urea (28876.298) were purchased from VWR. Fluorescent polystyrene particles of diameter 0.2 μm (FCDG003) were purchased from Bangs Laboratories. Poly-dimethylsiloxane (PDMS) SYLGARD 184 kits were purchased from Dow Corning. All the water used is Milli-Q. Imaging is done on a Nikon Ti2 Eclipse confocal microscope equipped with a Nikon DS-Qi2 camera using 20x air, 40x air, and 60x water objectives. The acquisition is done through the NIS-Elements software.

Polyethylene glycol diacrylate 12,000 Da (PEGDA 12k) was synthesized by reacting PEG 12k with acryolyl chloride[54]. Briefly, in a Schlenk flask flushed with nitrogen 10 g of PEG 12k, 68.5 mL of dry dichloromethane, 0.46 mL of triethylamine, and 0.35 mL of acryolyl chloride are left to react overnight while stirring. About half of the content of the flask was then evaporated (Rotovapor R-300 Büchi at 50 °C/690 mbar) and the rest was precipitated into ethyl ether. The powder was then vacuum-filtered.

### Condensate preparation
We prepare stock solutions with water: PEG 4k 60%w/v, KCl 4 M, urea 8 M, ammonia 5 M, sodium bicarbonate 1 M, NaOH 3 M, HCl 1 M, potassium phosphate buffer (KP) 0.5 M at pH = 7 (adjusted with pH meter Orion Star A111, Thermo Fisher). BSA and urease stock solutions concentration are measured after their preparation using a UV-vis spectrophotometer (Cary 60 Spectrophotometer, Agilent Technologies). Briefly, different dilutions in water are prepared (typically 200-100-50 for BSA and 15-10 for urease) and their absorbance spectra are measured in a quartz cuvette (UQ-124, Portmann Instruments). Using the extinction coefficients at 280 nm $\epsilon_{BSA}$ = 43.824 mM$^{-1}$cm$^{-1}$ and $\epsilon_{urease}$ = 54.165 mM$^{-1}$cm$^{-1}$ we calculate the concentration of each sample and average them to get the stock solution concentration (around 4.5 mM for BSA and 110 μM for urease). PEG 4k, BSA, urea, and urease stocks are stored at 4 °C.

Condensates are prepared by mixing the appropriate amount of stock solutions and water in a microcentrifuge tube. The total volume for the final solution is typically 1 mL, containing main components at overall concentrations of 23%w/v for PEG 4k, 3%w/v for BSA, 200 mM for KCl, and 100 mM for KP (pH 7). The concentrations of the extra components (urease, urea, NH$_3$ (aq), NaHCO$_3$ and NaOH) are variable as indicated in the main text. When needed, 0.2 μm fluorescent particles at a final dilution of ∼ 20000 (488 nm excitation, 525 nm emission filter) and Rhodamine-B at 10 μg/mL (561 nm excitation, 600 nm emission filter) are used for visualization. The particles are used for visualizing internal flows and rhodamine is used for visualizing condensates since it strongly partitions inside them. The depletant PEG 4k is always added as the very last component and the solutions are then gently mixed with the micropipette until homogeneous. The final volume fraction of condensates is of the order of $\phi_d = 3 \pm 2\%$ in the absence of extra components.

### Coatings preparation
The protocols for coating coverslips with PEGDA hydrogels is adapted from[25,26]. First, we silanize the glass coverslips. This is done by washing

the coverslips (VWR, #1 thickness) with water, ethanol, and again with water. We then expose them to UV-ozone (ProCleaner, Bioforce Nanosciences) for ~10 min. Then, ~50 μL of 95% v/v ethanol-water solution containing 0.3% v/v of the silane coupling agent 3-(Tri-methoxysilyl)propylmethacrylate are spread on the clean coverslips and left to react for 3 min. Finally, the reaction is quenched with ethanol and the coveslips are dried and stored with desiccant.

Subsequently, the silanized coverslip is coated with PEGDA solution containing photoinitiator and cured under UV light. Specifically, we dissolve the photoinitiator 2-Hydroxy-4′-(2-hydroxyyethoxy)-2-methylpropiophenone in water (1%w/v for PEGDA 700 or 4%w/v for PEGDA 12k) by sonicating in a bath sonicator for 30 min at 55 °C. The initiator solution is then added to pure PEGDA 700 solution at a 80/20 (v/v) ratio (or to a 40%w/v aqueous solution of PEGDA 12k at a 50/50 (v/v) ratio). We wash a fresh cover slip with water, ethanol, and again with water, and treat it with a commercial hydrophobic coating (Rain-X®). About 50 μL of the PEGDA and photoinitiator solution is then added to a silanized coverslip and then covered with a hydrophobic one. The glass-PEGDA 700 sandwich is then cured under 360 nm UV light for 20 min (or 1h for PEGDA 12k) and opened with a scalpel for immediate use, or stored in moist conditions. Our PEG-BSA condensates have a contact angle of $\approx 20°$ on bare glass, $\approx 100°$ on PEGDA 700 and $\approx 180°$ on PEGDA 12k[26].

### Active condensate experiments

We prepare three different condensate solutions with Rhodamine: solution 1 does not contain any additives, solution 2 contains urease ($c_e = 0.6$ to $1.2\,\mu M$) and fluorescent particles, and solution 3 contains urea (200 mM). Solutions 1 and 3 are then centrifuged until the dense and dilute phases are separated (16,000 g, 30 min), and the dilute phases are subsequently extracted and transferred to separate micro-centrifuge tubes. In the meantime, solution 2 is kept in a rotator and a PEGDA coated cover slip is prepared. If PEGDA 12k is used, the coating is equilibrated for 30 min with the previously extracted dilute phase of solution 1. The PEGDA-coated coverslip is then inserted into a Chamlide sample chamber (CM-S18-1, CM-S18-4, from Live Cell Instrument) and the chamber is filled with the dilute phase of solution 1 and a small amount of solution 2 (~1 μL). At this point, the sample chamber contains a a very dilute solution of urease-loaded protein condensates without urea. The chemical reaction is triggered by injecting an equal amount of the previously extracted dilute phase of solution 3, which contains urea, and monitored under the confocal microscope.

### Gradient chamber experiments

The gradient chambers are made of PDMS (cured overnight at 40 °C with a 10:1 ratio) pressed on a cover slip with a PEGDA 12k coating on the bridge location (see Fig. 2a). The PDMS mold is 3d printed (Pursa SL1) and thoroughly washed and dried before use. The dimensions of the reservoirs are $22 \times 5 \times 1\,mm$, the bridge is made out of a central stripe of $30 \times 1.5 \times 0.2\,mm$ and connecting teeth of $1.5 \times 1 \times 0.2\,mm$. The total distance between the reservoirs is thus 3.5 mm. Upon demolding, 2.5 to 3 mm holes are made with a hole puncher to allow fluid injection (circles in Fig. 2a) and the PDMS is gently pressed on a PEGDA coated cover slip to complete the chamber.

As in the active condensate experiment, we prepare three different passive condensate solutions with Rhodamine: solution 1 does not contain any additives, solution 2 contains fluorescent particles, and solution 3 contains the desired chemical (urea, $NH_3$ (aq), $NaHCO_3$, or NaOH). Solutions 1 and 3 are centrifuged and the dilute phase is extracted and transferred to clean microcentrifuge tubes. Solution 2 is diluted with the dilute phase of solution 1 (usually 1/3 (v/v)) and pipetted into the bridge openings. Capillarity then sucks the fluid in the whole bridge. Solution 1 is then gently pipetted into the first reservoir while capillarity prevents the invasion of liquid in the second reservoir. The chamber is then placed in a custom humidity box filled

with a wet sponge and placed under the microscope. Solution 3 is then slowly pipetted into the second reservoir until complete filling. There is some mixing in the bridge at this stage, making the initial conditions unknown, except when pH dye is used. Removing Rhodamine from solution 3 allows us to qualitatively gauge injection-induced mixing. The gradient chamber is left for > 1 h to equilibrate before any measurements are taken (except for the experiment of Fig. 3a, b involving a moving front).

When using a high concentration of $NH_3$ (aq) or NaOH, such that $pH \gtrsim 9$, solution 3 did not phase-separate and was injected as is. Adding the equivalent amount of $NH_3$ or NaOH directly to the dilute phase rather than in the condensate recipe did not yield significant differences, most likely because our overall composition is very close to the dilute phase composition (see Fig. 3c).

### pH measurements

pH measurements are performed either with active condensates or in the gradient chamber with passive ones. The procedure is very similar to what is described above for producing and visualizing droplets, except that instead of Rhodamine, the pH-sensitive dye HPTS is used at 400 μM final concentration. The pH measurment requires a calibration curve that relates the ratiometric fluorescence intensities (see below) to the pH. To this end, six to eight 200 μL calibration solutions are prepared for each experiment by mixing the dilute phases extracted from solution 1 and 3 at equal volumes and altering their pH in the range 7–9 by adding a few μL of NaOH or HCL (measured with Orion Star A111 pH meter). The calibration curve is sensitive to experimental conditions and should therefore be measured under identical conditions as the experiment: in Chamlide CM-S18-4 sample chambers with a PEGDA-coated cover slip, using the same microscope objective (usually 60x water) and performing a z-scan above the coating until fluorescence becomes independant of z.

The pH is determined from the ratios of the fluorescence intensities emitted at 525 nm for two different excitation wavelengths, 405 nm and 488 nm. The laser power and exposure are set to yield a signal-to-noise ratio above 10 at all pH values while care is taken to minimize sample heating. The raw signals are corrected for vignetting and the sensor noise (intensity with laser off) is subtracted. The calibration is performed at multiple locations in each sample chamber and the whole image is averaged. Fig. 5a shows a typical calibration curve $I_{488}/I_{405} = f(pH)$ whose inverse function $pH = f^{-1}(I_{488}/I_{405})$ is fitted by a third order polynomial. This fit is then used to convert the measured signal to pH values in the experiment. As shown in Fig. 5b, HPTS doesn't enter the droplets, and therefore the ratiometric signal is insufficient to extract meaningful pH values inside droplets. To measure $dpH/dx$ around satellite drops in active experiments, we extract the pH on a line aligned with the drop swimming direction and fit the pH around the drop (see Fig. 5c, d). For passive gradient chamber experiments, we average the pH on the width of the image (see Fig. 5e, f) and then fit a line around the drop position.

### Reaction diffusion model and numerical simulation

To understand the dependence of concentration profiles on droplet size, we need to account for coupled reaction and diffusion of multiple species in and around our droplet micro-reactors. To that end, we built a multi-component numerical reaction-diffusion model including the reactions of the enzyme with the substrate and its product with the buffer[55].

The system of chemical reactions for our active condensate system is the following (ignoring $PO_4^{3-}$ and $CO_3^{2-}$, which are negligible at the experimentally observed pH):

$$CO(NH_2)_2 + E \underset{K_m}{\rightleftharpoons} E - CO(NH_2)_2 \xrightarrow[H_2O]{k_{cat}} 2NH_3 + CO_2 \qquad (3)$$

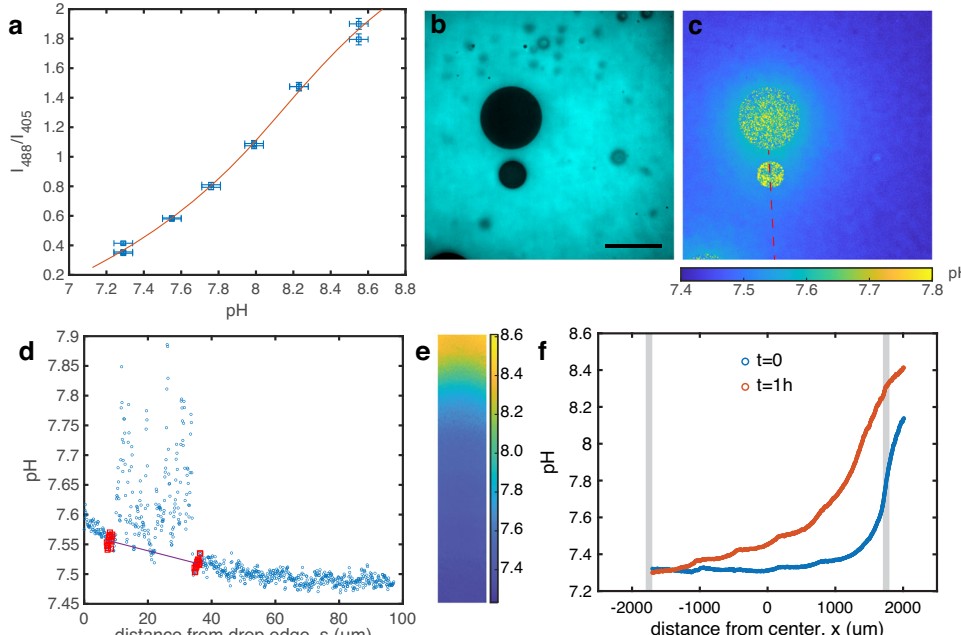

**Fig. 5 | pH imaging. a** Calibration curve for the pH measurements using HPTS. The solid curve is a cubic fit of the inverse function pH = f⁻¹($I_{488}/I_{405}$). Horizontal error bars represent the pH meter accuracy, vertical error bars represent the STD of fluorescence intensity over the whole calibration image. **b** Raw fluorescence image after excitation at 405 nm. The dye does not enter the droplets. Scale bar 50 μm, droplets are pinned on PEGDA 700. **c** Same image after conversion of the ratiometric fluorescence values to pH. The pH along the red dashed line is extracted and analyzed to determine the pH gradient around the lower satellite droplet. **d** pH profile from (**c**). 15 points around each edge of the droplets (red) are used to measure the pH gradient (line). **e** pH imaging in gradient chamber experiments. The pH in the whole bridge is shown 1 h after the initial injection of dilute phase supplemented with 60 mM of NH₃ (aq). **f** pH profile from e averaged along the width immediately after filling the reservoir and 1 h after. Gray bands indicate the edges of the bridge Source data for (**a**, **d**, **f**) are provided as a Source Data file.

Here a Michaelis-Menten kinetics with parameters $k_{cat}$ and $K_m$ is assumed for the enzymatic reaction and $k_i^{\pm}$ are the reactions rates of the proton exchange reactions.

Assuming no advection (i.e., Pe* = $\frac{VR}{D} \ll 1$), steady state (i.e., $\partial c/\partial t = 0$), and denoting $c_i$ the concentration of the component $i$ with CO(NH₂)₂:1, NH₃:2, NH₄⁺:3, CO₂:4, HCO₃⁻:5, H⁺:6, OH⁻:7, HPO₄²⁻:8, H₂PO₄⁻:9 the system of reactive transport equations is:

$$NH_4^+ \underset{k_2^-}{\overset{k_2^+}{\rightleftharpoons}} NH_3 + H^+ \tag{4}$$

$$CO_2 + H_2O \underset{k_3^-}{\overset{k_3^+}{\rightleftharpoons}} HCO_3^- + H^+ \tag{5}$$

$$H_2O \underset{k_4^-}{\overset{k_4^+}{\rightleftharpoons}} OH^- + H^+ \tag{6}$$

$$H_2PO_4^- \underset{k_5^-}{\overset{k_5^+}{\rightleftharpoons}} HPO_4^{2-} + H^+ \tag{7}$$

$$\nabla \cdot (-D_1 \nabla c_1) = -k_{cat} c_e \frac{c_1}{K_m + c_1} \tag{8}$$

$$\nabla \cdot (-D_2 \nabla c_2) = 2 k_{cat} c_e \frac{c_1}{K_m + c_1} + k_2^+ c_3 - k_2^- c_2 c_6 \tag{9}$$

$$\nabla \cdot (-D_3 \nabla c_3) = -k_2^+ c_3 + k_2^- c_2 c_6 \tag{10}$$

$$\nabla \cdot (-D_4 \nabla c_4) = k_{cat} c_e \frac{c_1}{K_m + c_1} - k_3^+ c_4 + k_3^- c_5 c_6 \tag{11}$$

$$\nabla \cdot (-D_5 \nabla c_5) = k_3^+ c_4 - k_3^- c_5 c_6 \tag{12}$$

$$\nabla \cdot (-D_6 \nabla c_6) = k_2^+ c_3 - k_2^- c_2 c_6 + k_3^+ c_4 - k_3^- c_5 c_6 + \tag{13}$$

$$k_4^+ - k_4^- c_6 c_7 + k_5^+ c_9 - k_5^- c_6 c_8 \tag{14}$$

$$\nabla \cdot (-D_7 \nabla c_7) = k_4^+ - k_4^- c_6 c_7 \tag{15}$$

$$\nabla \cdot (-D_8 \nabla c_8) = k_5^+ c_9 - k_5^- c_6 c_8 \tag{16}$$

$$\nabla \cdot (-D_9 \nabla c_9) = -k_5^+ c_9 + k_5^- c_6 c_8. \tag{17}$$

Now following the steps of ref. 55, we rearrange Eqs. (8)–(17) in order to eliminate the unknowns $k_i^{\pm}$:

$$\nabla \cdot (-D_1 \nabla c_1) = -k_{cat} c_e \frac{c_1}{K_m + c_1} \tag{18}$$

$$\nabla \cdot (-D_2 \nabla c_2 - D_3 \nabla c_3 - 2 D_1 \nabla c_1) = 0 \tag{19}$$

$$\nabla \cdot (-D_4 \nabla c_4 - D_5 \nabla c_5 - D_1 \nabla c_1) = 0 \tag{20}$$

$$\nabla \cdot (-D_6 \nabla c_6 - D_3 \nabla c_3 + D_5 \nabla c_5 + D_7 \nabla c_7 + D_8 \nabla c_8) = 0 \tag{21}$$

**Table 1 | Parameters for the simulation of eqs. (18)–(22) and (23)–(26)**

| | CO(NH₂)₂ | NH₃ | NH₄⁺ | CO₂ | HCO₃⁻ | H⁺ | OH⁻ | HPO₄²⁻ | H₂PO₄⁻ |
|---|---|---|---|---|---|---|---|---|---|
| $D_i^w$ (10⁻⁹ m²/s) | 1.38 | 1.50 | 1.96 | 1.67 | 1.19 | 9.31 | 5.27 | 0.76 | 0.96 |
| $c_i^\infty$ (mM) | $c_1^\infty$ | 0 | 0 | 0 | 0 | $10^{-pH^\infty}$ | $K_w/10^{-pH^\infty}$ | $100/(1+\frac{c_6^\infty}{Ka_5})$ | $\frac{c_6^\infty c_8^\infty}{Ka_5}$ |
| | $K_m = 23$ mM | pKa₂ = 9.25 | | pKa₃ = 6.35 | | pK$_w$ = 14 | | pKa₅ = 7.21 | |

Diffusion coefficient and pK are extracted from ref. 62, urea-urease Michaelis-Menten constant from ref. 19.

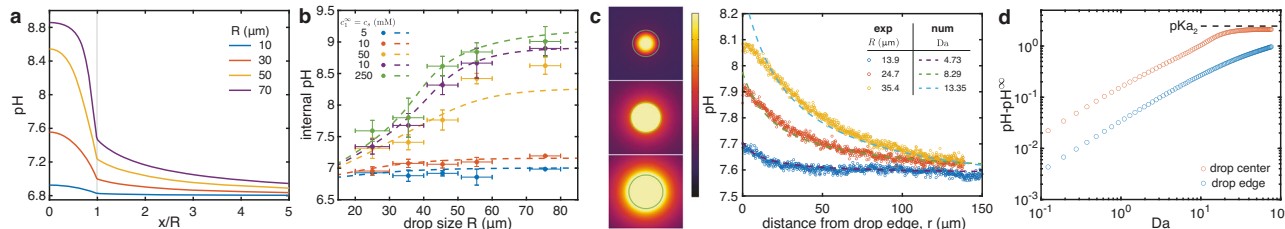

**Fig. 6 | Reaction-diffusion simulations. a** pH profiles as a function of the dimensionless distance $x/R$ as the drop radius $R$ increases (pH$^\infty$ = 6.8, $\eta_{dense}$ = 0.8 Pa s, $c_s = c_1^\infty = 100$ mM, $c_e^d = 1\mu$M, $k_{cat} = 4400$ 1/s). **b** Internal pH as a function of the drop radius $R$ for various urea concentrations $c_s$. Data extracted from ref. 19, circles are mean values and vertical error bar the SD of the internal pH measured in 5 independent droplets. Horizontal error bars represent the edges of the bin used to bin the data in ref. 19. Dashed lines are simulations (same parameters as a, except for $c_s$ that is varied). **c** Numerical prediction for the pH around the reacting droplets shown in Fig. 1f ($c_e^d = 40\mu$M, $k_{cat} = 352$ 1/s, pH$^\infty$ = {7.59, 7.59, 7.54} and $\eta_{dense}$ = {0.8, 0.3, 0.2} Pa s). Images of the numerical simulations are shown on the left (the green circles indicates the drop edge), and pH profiles outside of the drop extracted from the experiments and simulations are shown on the right. **d** Increase in pH due to the activity pH − pH$^\infty$ as a function of the Damköhler numbers Da in the drop center and at the edge (same parameters as a). The black dashed line is pH = pKa₂. Source data for (**a**–**d**) are provided as a Source Data file.

$$\nabla \cdot \left(-D_8 \nabla c_8 - D_9 \nabla c_9\right) = 0. \tag{22}$$

This yields 5 equations with 9 unknowns, and we obtain the 4 missing equations by assuming equilibrium of the proton exchange reaction:

$$Ka_2 = \frac{k_2^+}{k_2^-} = \frac{c_2 c_6}{c_3}, \tag{23}$$

$$Ka_3 = \frac{k_3^+}{k_3^-} = \frac{c_5 c_6}{c_4}, \tag{24}$$

$$Ka_4 = \frac{k_4^+}{k_4^-} = K_w = c_6 c_7, \tag{25}$$

$$Ka_5 = \frac{k_5^+}{k_5^-} = \frac{c_8 c_6}{c_9}. \tag{26}$$

Eqs. (18)–(22) and (23)–(26) are solved numerically for a spherically symmetric droplet (neglecting the surface) with the COMSOL Multiphysics® software, assuming no flux at the origin $\partial c_i/\partial r|_{r=0} = 0$ and constant concentrations at infinity $c_i(\infty) = c_i^\infty$. We assume no partitionning for all small molecules ($c_i(R^-) = c_i(R^+)$) and infinite partitioning for the enzyme ($c_e(r<R) = c_e^d$, and $c_e(r>R) = 0$). This discontinuity is enforced numerically with the smooth step function $S(r) = \frac{1}{2}\left(1 - \tanh\left(100\frac{r-R}{R}\right)\right)$ and the mesh is tightened around $r = R$. Similarly we expect a discontinuity of all the diffusivities $D_i$ at the condensate edge due to large viscosity contrast between the dense and dilutes phases. Since our diffusing components are much smaller than the macromolecules in both the dense and dilute phases, obstruction theory[56] rather than the Stokes-Einstein relationship will govern their diffusivity. Following ref. 57,

we estimate the diffusivities in the dense and dilute phases using their respective viscosities: $D_i = D_i^w (\eta/\eta^w)^{-3/5}$ with $D_i^w$ the diffusivity of species $i$ in water and $\eta^w = 10^{-3}$ Pa s the viscosity of water. We impose the diffusivity discontinuity numerically using the same smooth step function S(r).

Table 1 shows the values of parameters kept constant in all the simulations. We further use $\eta_{dilute} = 24$ mPa s for the dilute phase[19] in all simulations. The concentration of enzymes in the drop is estimated assuming infinite partitioning, i.e., $c_e^d = c_e/\phi_d$. Since the volume fraction of condensate of $\phi_d = 3 \pm 2\%$ can vary substantially there is a large uncertainty on $c_e^d$. In the far-field the experimental values of urea concentration $c_1^\infty = c_s$ and pH$^\infty$ = − log($c_6^\infty$) are used while the viscosity inside the droplet $\eta_{dense}$ is adjusted within the experimentally measured range 0.1–1 Pa s (see values for each simulation below). The catalytic constant $k_{cat}$ depends on the activity of the urease used (which can decrease as the enzyme ages) as well as the experimental conditions (buffer, pH, temperature). For jack beans urease we expect $k_{cat} \sim 10^3$–$10^4$ 1/s[19,58]. Given that the experimental value of both $c_e^d$ and $k_{cat}$ are uncertain and that they act together in eq. (18), we use $k_{cat}$ as fitting parameter.

We first show in Fig. 6a typical numerical pH profiles as we vary the drop size $R$. The pH is larger inside the drop and decreases close to the edge where it exhibits a slope discontinuity; the overall pH magnitude and extent increases with the size of the drop as seen in experiments. We then compare our simulations to experiments. Figure 6b shows the simulated internal droplet pH and the one measured in ref. 19. We find a quantitative agreement with a reasonable value of the catalytic activity $k_{cat} = 4400$ 1/s (all other parameters are known). Finally, we compare our simulation to the data of Fig. 1f for the pH outside of the droplets in Fig. 6c. We find a quantitative agreement for the three drops with a single value of $k_{cat} = 352$ 1/s. Our model thus captures the radius-dependence of the pH for the three droplet sizes shown in Fig. 1f, $R$ = {13.9, 24.7, 35.4} μm. The remaining parameters of the model are $c_e^d = 40\mu$M, pH$^\infty$ = {7.59, 7.59, 7.54} and $\eta_{dense}$ = {0.8, 0.3, 0.2} Pa s. The $k_{cat}$ is lower than expected, which suggests that our enzyme batch may have lost some of its activity.

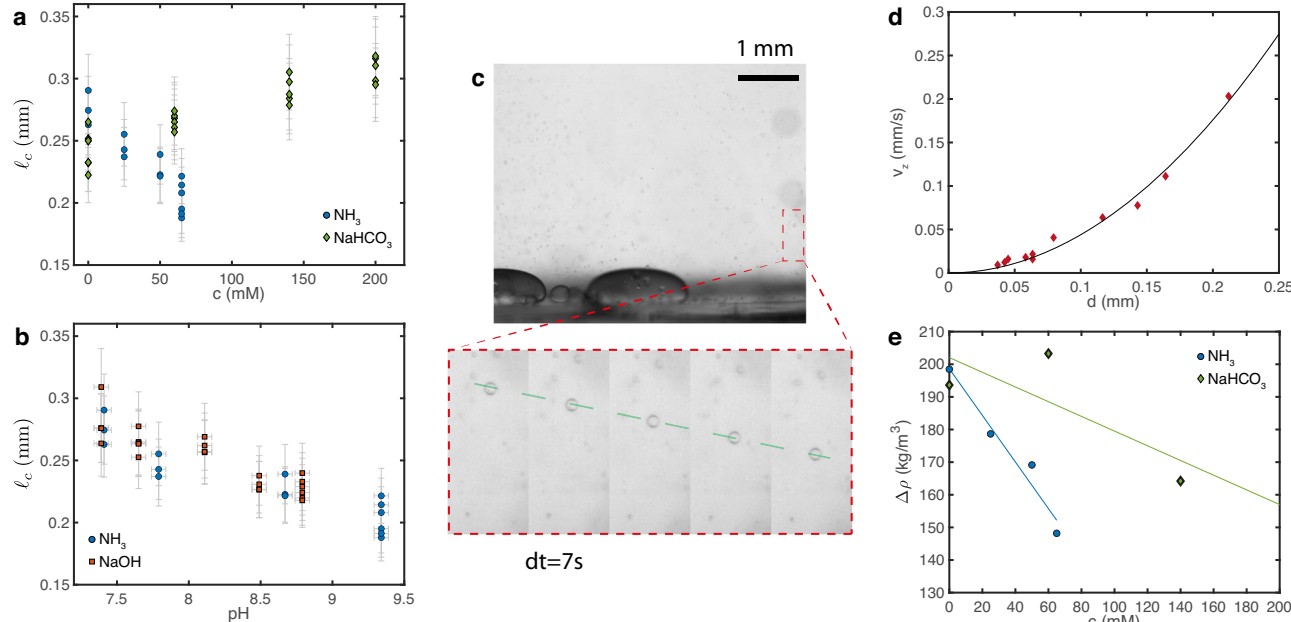

**Fig. 7 | Interfacial tension measurment. a, b** Capillary length $\ell_c$ as a function of solute concentration $c$ and pH for $NH_3$ (aq), $NaHCO_3$ and NaOH. Each point represents a measurement on one droplet. Error bars convey non-statistical measurement uncertainties that account for the accuracy of the measuring instruments and measuring procedure (see text for details) **c** Snapshot of a movie looking at the sedimentation of small drops. Scale bar 1 mm, PEGDA 12k coating. The inset shows an image sequence zoomed on a single drop ($dt$ = 7 s). The drop has reached a constant velocity as shown by the dashed line. **d** Terminal velocity $v_z$ as a function of the drop diameter $d$. The solid curve is a quadratic fit $v_z = Ad^2$, from which we extract the density difference, $\Delta\rho$. **e** Density difference $\Delta\rho$ as a function of solute concentration $c$ for $NH_3$ (aq) and $NaHCO_3$. The solid lines are linear fits that we use to compute $\gamma$, the distance to the fitted lines is our measurement error $\delta(\Delta\rho)$. Source data for (**a**, **b**, **d**, **e**) are provided as a Source Data file.

To understand this radius-dependence, we come back to Eqs. (18)–(22). Since the pH increase is driven by the enzymatic reaction inside the droplet (where $D_i$ are constants), we isolate it and make it dimensionless using $\tilde{r} = r/R$ and $\tilde{c}_1 = c_1/K_m$ :

$$\tilde{\nabla}^2 \tilde{c}_1 = \mathrm{Da}\frac{\tilde{c}_1}{1+\tilde{c}_1}, \tag{27}$$

$$\mathrm{Da} = \frac{k_{cat}c_e^d R^2}{D_1 K_m}. \tag{28}$$

This equation is governed by a single dimensionless number, the Damköhler number Da, which compares the diffusion and reaction timescales. If Da $\ll$ 1, diffusion is much faster than the reaction and the latter becomes unnoticeable (no pH increase). The reaction speed becomes significant only when Da $\gtrsim$ 1. Since Da ~ $R^2$, the drop size has a strong influence on the transition between the diffusion- and reaction-dominated regimes. Figure 6d shows the simulated pH increase as a function of Da inside and at the edge of the drop. The pH increases noticeably (by more than 0.1) only for Da > 1. The pH increase eventually stalls when the system reaches the pKa of $NH_3$, which effectively becomes the dominant buffer of the system, a feature also present in the experiment (Fig. 6b). Finally, while for Da $\ll$ 1 the substrate has time to diffuse inside the droplet before being consumed, for Da $\gg$ 1, the reaction rate is so fast that it is consumed as soon as it reaches the drop edge. We therefore expect the total rate of products released to scale with the volume of the condensate in the first case but with the surface area of the condensate in the latter.

### Interfacial tension measurement

We use the sessile drop method to measure the interfacial tension between the dilute and dense phases[19,59]. We prepare large samples (~2 mL) of passive PEG-BSA condensates at various concentrations of $NH_3$ (aq), $NaHCO_3$ and NaOH. We then centrifuge each sample,

separate the dense and dilute phases, and measure the dilute phase pH. In the meantime, we cut a glass cover slip into 8 × 8 mm pieces and coat them with PEGDA 12k (see procedure above). The coated glass is then inserted into a standard quartz cuvette that is then filled with dilute phase. We then pipette ≈ 0.5–1 μL of dense phase at multiple locations in the cuvette to form large condensates as shown in the inset of Fig. 2e. These drops are large enough to be flattened by gravity and we can measure their capillary length $\ell_c = \sqrt{\gamma/(\Delta\rho g)}$ by fitting their shape to the Young-Laplace equation. The experimental setup and fitting code are described in ref. 59.

The capillary lengths of individual drops are shown in Fig. 7a, b. To calculate the interfacial tension $\gamma = \Delta\rho g \ell_c^2$, we then measure the density difference between the dense and dilute phases, $\Delta\rho$, through the sedimentation velocity of drops of various sizes. We generate clouds of small droplets by mixing the content of the cuvette with a pipette. After some time the background flow vanishes and we record the sedimentation velocity $v_z$ of droplets of different sizes $d$ as shown in Fig. 7c. For isolated drops, assuming that they have reached their terminal velocity and that $\eta_{dense} \gg \eta_{dilute}$, we extract the density difference from a quadratic fit of the data: $v_z = \Delta\rho g d^2/(18\eta_{dilute})$ (see Fig. 7d). We plot the extracted density difference in Fig. 7e. Given the large variability, we fit the measured $\Delta\rho$ with a line and use the fit value to calculate $\gamma$ in Fig. 2e, f. The vertical error bars in Fig. 2e, f combine measurement errors on both $\Delta\rho$ and $\ell_c$, $\frac{\delta\gamma}{\gamma} = \sqrt{\left(\frac{\delta(\Delta\rho)}{\Delta\rho}\right)^2 + 2\left(\frac{\delta\ell_c}{\ell_c}\right)^2}$, with $\delta\ell_c = 0.1\ell_c$ and $\delta(\Delta\rho)$ the distance between the measured value and the fitted value in Fig. 7e. The pH meter measurement accuracy $\delta$(pH) = 0.05 is used as the horizontal error bar in Fig. 2f, g.

### Viscosity measurement

We use passive microrheology to measure the viscosity in the dense phase. For each concentration of $NH_3$ (aq) and NaOH, we deposit a diluted passive condensate solution in a sealed sample chamber.

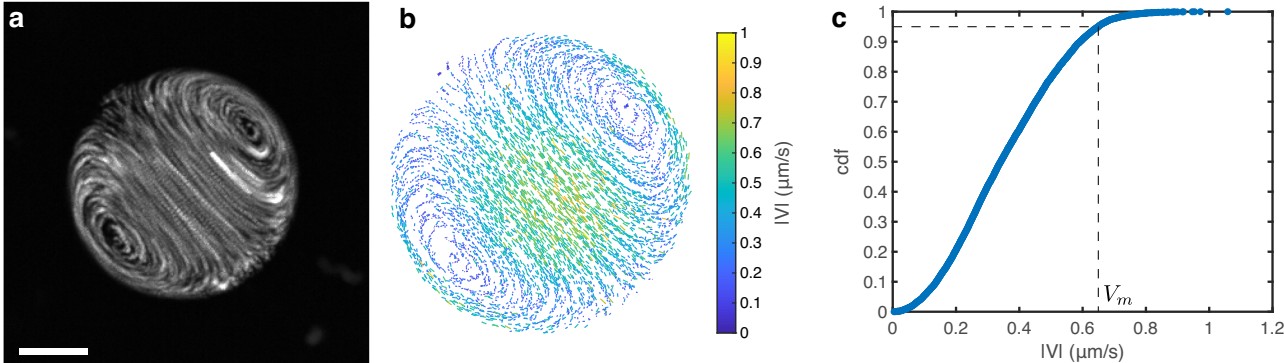

**Fig. 8 | Particle tracking velocimetry. a** Time projection of fluorescent particles inside a satellite droplet pinned on a PEGDA 700 coating. Scale bar 10 µm. **b** Quiver plot after particle tracking for the droplet shown in (**a**). The color codes the magnitude of the internal velocity |$V$|, only half of the tracks are displayed for clarity. **c** Cumulative distribution function of the internal velocity for the same droplet. Since the highest recorded values are likely to be noise from tracking errors, we define the maximum internal velocity $V_m$ as the velocity larger than 95% of our tracked velocities as shown by the dashed line. Source data for (**c**) are provided as a Source Data file.

Specifically, for each concentration we prepare two solutions of passive PEG-BSA condensate: solution 1 contains fluorescent particles (diameter 0.2 µm) while solution 2 doesn't. We centrifuge solution 2 and separate the two phases. In the meantime, we prepare a cover slip with a PEGDA 700 coating. We make several sample chambers by pressing an Invitrogen™ Press-to-Seal™ silicone isolator on the coated cover slip. For each concentration, we deposit ~20 µL of the dilute phase extracted from solution 2 in a chamber and then a small amount of solution 1 (~0.2 µL). We then seal the chambers by pressing a clean cover slip on top. The sealing must be excellent to avoid evaporation, which drives flows that hinder the particle diffusivity measurement.

The motion of the particles in a 2d plane is then recorded under the microscope with a 60x water objective. The recording framerate was adjusted between 5 fps and 0.5 fps depending on the sample viscosity. Particle tracking is done with a custom Matlab code available at https://doi.org/10.5281/zenodo.4884194. The particle diffusivity is extracted from the tracks using the optimal estimator from ref. 60 corrected for a constant background flow:

$$D = \frac{\overline{(\Delta x_n)^2}}{2\Delta t} + \frac{\overline{\Delta x_n \Delta x_{n+1}}}{\Delta t} - \frac{3}{2}\frac{\overline{\Delta x_{n+1}\Delta x_{n-1}}}{\Delta t}. \quad (29)$$

Here, $\overline{(\cdot)}$ denotes an average over all tracks (all particles, all time points, both x and y direction), $\Delta x_n$ the displacements of a tracked particle between frames $n+1$ and $n$ and $\Delta t$ the time between two frames. We further checked that the estimated diffusivity agreed with the mean squared displacement after correcting the tracks for the average background flow of the experiment, if present. The viscosity is then calculated from the Stokes-Einstein relationship $\eta = kT/(6\pi a D)$ with $k$ the Boltzmann constant, $T$ the absolute temperature and $a = 0.1 \,\mu m$ the particle radius. This measurement is performed for several individual droplets in a given sample. The vertical error bars in Fig. 2g represent the measurement errors on $D$ : $\frac{\delta \eta}{\eta} = \frac{\delta D}{D}$, set to 20%.

**Internal velocity measurement**
The internal velocity of drops pinned on the surface is measured through particle tracking velocimetry in experiments where pH gradients are recorded (using the 60x water objective, see Fig. 8a). The tracking is done in a 2d plane close to the drops center with the same in-house MATLAB® code as the microrheology experiments. By differentiating the tracks we get the instantaneous velocity of each tracked particle as shown in Fig. 8b for a satellite drop in an active experiment. We recover the characteristic internal flow of Marangoni swimmers with its pair of vortices. Theory for Marangoni swimmers in infinite space subject to a linear interfacial tension gradient yields simple predictions for the swimming velocity and maximum internal velocity (reached at the drop center and edge, as shown in Fig. 8b)[34]. Since we use pinned drops for pH measurements (that do not swim), we compare the maximum internal velocity to the theoretical prediction. To measure the maximum velocity in the presence of some tracking error (the fastest recorded velocities are likely to be noise), we compute the cumulative distribution function of the velocity magnitude |$V$| shown in Fig. 8c. We define the maximum internal velocity |$V_m$| as the velocity greater than 95% of all the recorded velocities, i.e., the velocity for which the cumulative distribution function is 0.95. The vertical error bar in Fig. 2g is a measurement error (set to 20%), while the horizontal error bar is a combination of measurement errors on $dy/d$pH and $d$pH/$dx$.

**Critical point measurement**
To measure the critical point of our BSA-PEG system, we first prepared a suspension with the highest possible volume fraction of droplets given our stock solutions. Specifically, we used 17%w/v of PEG 4k, 13% w/v of BSA, 200 mM of KCl, 100 mM of KP (pH 7) to make the droplets. And to see the effect of additional compounds, either 50 mM of $NH_3$ (aq) or 200 mM of $NaHCO_3$ were added. The suspension was then centrifuged and equal volumes of dense phase and dilute phase were transferred to an empty microcentrifuge tube, resulting in 50% volume fraction of droplets and placing us at the center of the tie-line. The resulting solution was then diluted with buffer containing just 200 mM of KCl and 100 mM of KP (pH 7) and the additional components (50 mM $NH_3$ (aq) or 200 mM $NaHCO_3$), if any. Upon dilution, the droplet suspension was gently mixed and centrifuged to see whether the solution was still phase-separated. If so, we checked whether the volume fraction of dense phase remained at 50% and if not, the volumes of the two phases were adjusted to achieve 50% volume fraction. This dilution process was repeated in small increments until the solution no longer phase separated. At that point, the composition of this final solution was analyzed by measuring the concentration of BSA with UV-vis spectroscopy (see above) and the concentration of PEG 4k with NMR spectroscopy[19,61]. Briefly, 50 µL of the solution and 30 µL of DMF were dispersed in 570 µL of $D_2O$ and placed in Schott NMR sample tubes. ${}^1$H NMR spectra were acquired using a 300 MHz instrument (300 Ultrashield, Bruker) and the resulting data were analyzed using the MestreNova software (Mestrelab Research). The PEG 4k concentration was determined by comparing the integrals of the two DMF methyl peaks around 3 ppm (6 protons per molecule) to the intensity of the PEG peak around 3.7 ppm (~354 protons per molecule).

## Statistics and reproducibility

Active droplet swimming on PEGDA 12k experiments such as the ones shown in Fig. 1c have been repeated 11 times independently with similar results to gather the data of Fig. 1e. Active droplet stuck on PEGDA 700 experiments for coupled internal velocity and pH gradient measurements such as the ones shown in Figs. 1b, 5b, and 8a and have been repeated 11 times independently with similar results to gather the data of Fig. 2h. Passive droplets in a gradient chambers experiments have been repeated 11 times independently with various gradient strengths for $NH_3$ with similar qualitative results (e.g., Fig. 2b), 7 times for NaOH (e.g., Fig. 2c), and 6 times for $NaHCO_3$ (e.g., Fig. 2d).

## Reporting summary

Further information on research design is available in the Nature Portfolio Reporting Summary linked to this article.

## Data availability

Source data are provided as a Source Data file. In addition, a representative sample of raw movies are available at https://doi.org/10.5281/zenodo.10912624. Source data are provided with this paper.

## Code availability

The codes used in this study are available at https://doi.org/10.5281/zenodo.10912624.

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

## Acknowledgements

We thank Magdalini Polymenidou for suggesting the term dialytaxis, Iacopo Mattich for help with 3d printing, and Gianna Wolfisberg for help with NMR measurements. This work was supported by grants from the Swiss National Science Foundation (National Center of Competence in Research Bio-Inspired Materials to E.R.D., grant number 172824 to E.R.D., and grant number 202214 to A.A.R.).

## Author contributions

E.J.P., A.T. and C.L. performed the experiments. E.J.P., R.W.S., A.A.R. and E.R.D. analyzed and interpreted the experiments. E.J.P. developed the reaction-diffusion model and simulations. E.R.D. conceived and supervised the project. E.J.P. and E.R.D. wrote the paper with inputs from all authors.

## Competing interests

The authors declare no competing interests.
