## [Peer Review File · Nature Communications]

Reviewers' Comments:

Reviewer #1:

Remarks to the Author:

I have considered the rebuttal by the authors. Overall they have done a fair job addressing my comments.

My main concern is whether the dialytaxis mechanism can win over other passive or active mechanisms, and their back of the envelope calculations suggest this is possible at least for some condensates. However, I note that they have used the higher end of interfacial tensions and lower end of viscosities reported in the literature. For example, see Wang et al. doi: 10.1016/j.bpr.2021.100011 for tabulated values. For the opposite limits of low surface tension and high viscosity, it is less clear. I suggest the authors to use a broader range of values in their estimates, but I agree with them now that the dialytaxis mechanism can be relevant for some condensates.

I also find it difficult to judge whether the paper is suitable for publication in Nature Communications when the article itself is not yet revised. Hence I will reserve that judgement until I see the revised manuscript. I would recommend the editors to invite the authors for revision. For example, I think the above-mentioned estimates should be included in the manuscript.

Reviewer #2:

Remarks to the Author:

The authors have helpfully answered several of my questions in their response letter. However, there were a number of comments to which the authors replied that "we will be happy to revise this schematic if the editors invite revision." It doesn't appear that the edits have been made yet. I would ask that the authors please make those revisions. My other questions have been addressed in the response.

**Response to the Reviewer's Comments for Manuscript NCOMMS-23-52222-T:
Phase-Separated Droplets Swim to Their Dissolution
by Etienne Jambon-Puillet, Andrea Testa, Charlotta M Lorenz, Robert W Style,
Aleksander Rebane, Eric Robert Dufresne**

Reviewer #1 (Remarks to the Author):

I have considered the rebuttal by the authors. Overall they have done a fair job addressing my comments.

My main concern is whether the dialytaxis mechanism can win over other passive or active mechanisms, and their back of the envelope calculations suggest this is possible at least for some condensates. However, I note that they have used the higher end of interfacial tensions and lower end of viscosities reported in the literature. For example, see Wang et al. doi: 10.1016/j.bpr.2021.100011 for tabulated values. For the opposite limits of low surface tension and high viscosity, it is less clear. I suggest the authors to use a broader range of values in their estimates, but I agree with them now that the dialytaxis mechanism can be relevant for some condensates.

I also find it difficult to judge whether the paper is suitable for publication in Nature Communications when the article itself is not yet revised. Hence I will reserve that judgement until I see the revised manuscript. I would recommend the editors to invite the authors for revision. For example, I think the above-mentioned estimates should be included in the manuscript.

We have now revised the manuscript to include a Peclet number estimation, now using a wider range of physical parameters as suggested.

“The strength of dialytaxis compared to diffusion can be estimated using a Peclet number $\mathrm{Pe} = UR/D$. Combining Eq.~\eqref{eq:vth} with Stokes-Einstein relationship for the condensate diffusivity, we have $\mathrm{Pe} \approx (\eta_{\text{dilute}}/\eta)(\pi R^2 \Delta\gamma/kT)$. The surface tension difference $\Delta\gamma$ depends on the solute concentration gradient and how much it perturbs the phase equilibrium but is bounded by γ , the condensate interfacial tension. Condensate physical properties can vary by at least an order of magnitude depending on the condensate, and are often only measured for simplified *in vitro* systems. Nonetheless, assuming a moderate perturbation of the phase equilibrium $\Delta\gamma \sim 0.1\gamma$ and using typical values found in the literature\cite{Brangwynne2009,feric2016coexisting,Boddeker2022,WANG2021,Jawerth2018} $10 < \eta/\eta_{\text{dilute}} < 100$, $0.1 < \gamma$; ($\mu\text{N/m}$) < 100 , and $0.5 < R$; (μm) < 10 we find $10^{-2} < \mathrm{Pe} < 10^5$. This simple estimate suggests that dialytaxis could be stronger than diffusion for a wide range of condensates, especially large ones, with small viscosity contrast, and a large interfacial tension.”

We further included the other referee's comments and suggestions (see below and revised manuscript).

We thank again the referee for their comments and questions which greatly improved our manuscript and hope it is now suitable for publication in Nature Communications.

Reviewer #2 (Remarks to the Author):

The authors have helpfully answered several of my questions in their response letter. However, there were a number of comments to which the authors replied that “we will be happy to revise this schematic if the editors invite revision.” It doesn't appear that the edits have been made yet. I would ask that the authors please make those revisions. My other questions have been addressed in the response.

We have now revised the manuscript as proposed in the previous response. Here are the original referee's remarks and our original response with the corresponding edits done in the revised manuscript:

1. Authors use the term “chemically active” page 1 line 35. I don't understand what this means. Do they mean that chemical gradients exist? That a reaction is occurring? There can be reactions and no gradients so I think this is worth clarifying.

We thank the referee for pointing this out, we implied that any reaction will produce a gradient which is incorrect. We will be happy to revise this sentence if the editors invite revision.

We have now changed “chemically active” page 1 line 35 to “generate chemical gradients”

2. The experimental design depicted in Fig 2A is not very clear. Caption refers to a top and bottom reservoir but the image looks like there are two side-by-side reservoirs. There are no labels. Please improve

We thank the referee for their feedback and we will be happy to revise this schematic if the editors invite revision.

We have now edited Fig 2a to add label and rewrote the caption for clarity

3. I understand why the authors find it convenient to coin a new term dialytaxis, but this is still the Marangoni effect. Dissolution is just one of many ways that interfacial tension gradients can be created. It would be good to emphasize more in the manuscript that this mechanism is indeed the Marangoni effect. For example the abstract does not make this point and is a bit misleading; it gives the impression that there is some new sort of “taxis” being discovered, which there is not.

We thank the referee for their suggestion. We will be happy to modify the abstract as suggested if the editors invite a revision.

We now mentioning the Marangoni effect in the abstract

4. "In stark contrast to previous examples of active Marangoni swimmers, however, NH₃ (aq), NaOH, and NaHCO₃ are not surfactants". This statement is too strong and should be modified or removed. It is obvious that surfactants are not a requirement to modify interfacial tension for droplets. They are simply convenient which is why they are often used. There are examples where no surfactant is used, for example: Mode selection in the spontaneous motion of an alcohol droplet, Ken Nagai, Yutaka Sumino, Hiroyuki Kitahata, and Kenichi Yoshikawa Phys. Rev. E 71, 06530, 2005.

We agree with the referee that Marangoni swimmers without surfactants have been reported. We will be happy to correct this error if the editors invite a revision.

We have replaced the sentence by "The strong effect of these solutes on the interfacial tension of our aqueous PEG-BSA system is, however, unexpected. They are not surfactants and barely affect the interfacial tension of water at these concentrations \cite{Rice:1928,BEATTIE:2014}"

5. This dialytaxis framework assumes there are no interfacially active chemical species present. However, it's not clear if this is actually the case within biological systems, and it is certainly not the case for oil-water systems with surfactant containing active droplets. Authors should discuss the limitations of their conclusions and conditions wherein this framework would not hold.

The referee is right, our analysis assumes no surface active chemicals. This is an important point, and we will be happy to highlight it if the editors invite revision.

We have added the two following sentences on p4 line 225 and 233

"We have introduced dialytaxis which has been rationalized in the context of macromolecular solutions, in the absence interfacially active chemical species. However, we note that it qualitatively describes the behavior of other two-phase systems."

"Dialytaxis may provide a simple unifying framework for a wide range of solutal Marangoni effects. In the presence of surface active molecules, however, it may not have a dominant effect."

We further added the discussion about the strength of dialytaxis compared to diffusion we had with the other referee (see above and revised manuscript).

We thank again the referee for their comments and corrections which greatly improved our manuscript and hope it is now suitable for publication in Nature Communications.

Reviewers' Comments:

Reviewer #1:

Remarks to the Author:

Overall I am mostly satisfied with the changes made by the authors.

However, I would like to see the issues related to the effects of solid substrate discussed in the final manuscript. These points seem to be lost between the previous and current revisions (see my initial review and the authors' response to the initial review).

After this change, I think the manuscript is suitable for publication in Nature Communications.

Reviewer #2:

Remarks to the Author:

Thank you for editing the manuscript in accordance with my suggestions. I think the paper is suitable for publication.

**Response to the Reviewer's Comments for Manuscript NCOMMS-23-52222-T:
Phase-Separated Droplets Swim to Their Dissolution
by Etienne Jambon-Puillet, Andrea Testa, Charlotta M Lorenz, Robert W Style,
Aleksander Rebane, Eric Robert Dufresne**

Reviewer #1 (Remarks to the Author):

Overall I am mostly satisfied with the changes made by the authors.

However, I would like to see the issues related to the effects of solid substrate discussed in the final manuscript. These points seem to be lost between the previous and current revisions (see my initial review and the authors' response to the initial review).

After this change, I think the manuscript is suitable for publication in Nature Communications.

We have now revised the manuscript to clarify the two substrates used in the study when describing the experimental system

“On a glass surface functionalized with polyethylene glycol diacrylate with $M_n=700$ (PEGDA 700), active condensates develop internal flows oriented toward neighboring droplets, but remain pinned (Fig.~\ref{fg:fig1}b) \cite{Testa2021}. Using a much larger molecular weight of $M_n=12\,000$ yields a swollen gel (PEGDA 12k) that substantially reduces the adhesion energy between droplets and surfaces (see \cite{spanke2020wrapping,testa2023} and Methods). On this surface, reacting condensates swim toward each other with speeds on the order of $\mu\text{m/s}$ (Fig.~\ref{fg:fig1}c, Movie~S1, and \cite{dindo2023chemotactic}).”

We further now explicitly mention in the main text and caption which coating is used for all experiment.

We now cite [10.1101/2023.07.03.547532](https://doi.org/10.1101/2023.07.03.547532) which appears very relevant to our study.

Other changes are made to comply with Nature Communications format and policies.

We thank again the referee for their comments and questions which greatly improved our manuscript and hope it is now suitable for publication in Nature Communications.

Reviewer #2 (Remarks to the Author):

Thank you for editing the manuscript in accordance with my suggestions. I think the paper is suitable for publication.

We thank the referee for their assessment deeming our manuscript suitable for publication in Nature Communications.

We have made some modification to the manuscript to answer reviewer 1 and comply with Nature Communications format and policies. We also added a citation (see reply to referee 1).

Note that we also slightly modified a sentence from the last report for clarity:

“We have introduced dialytaxis in the context of macromolecular solutions without interfacially active chemical species.”